# Chemo-Radio-Immunotherapy for NSCLC III: ESR/ATS Thresholds for DL_CO_ Correlate with Radiation Dosimetry and Pneumonitis Rate

**DOI:** 10.3390/cancers15071966

**Published:** 2023-03-25

**Authors:** Markus Stana, Brane Grambozov, Josef Karner, Isabella Gollner, Christoph Gaisberger, Elvis Ruznic, Barbara Zellinger, Raphaela Moosbrugger, Michael Studnicka, Gerd Fastner, Felix Sedlmayer, Franz Zehentmayr

**Affiliations:** 1Department of Radiation Oncology, Paracelsus Medical University, 5020 Salzburg, Austria; m.stana@salk.at (M.S.); b.grambozov@salk.at (B.G.); e.ruznic@salk.at (E.R.);; 2Institute of Pathology, Paracelsus Medical University, 5020 Salzburg, Austria; b.zellinger@salk.at; 3Department of Pulmonology, Paracelsus Medical University, 5020 Salzburg, Austria

**Keywords:** NSCLC stage III, durvalumab, immunotherapy, chemoradiotherapy, ESR/ATS thresholds, DL_CO_

## Abstract

**Simple Summary:**

Since the approval of durvalumab for the treatment of unresectable non-small-cell lung cancer UICC stage III, the 5-year overall survival rates have risen from below 20% to 50%. Although the validity of lung function testing has been questioned, for long-term survivors, residual pulmonary capacity after treatment is very important in terms of quality of life. The clinically most widely used lung function parameters are forced expiratory volume in one second (FEV_1_) and carbon monoxide diffusing capacity (DL_CO_). As the latter represents the alveolar compartment, it seems more suitable in the radiotherapy context. In the current analysis, we can show that DL_CO_ correlates with radiation dosimetry and the incidence of pneumonitis. Hence, from a clinical point of view, peri-treatment lung function testing is indispensable as it helps to optimize radiation treatment planning and predicts pulmonary toxicity.

**Abstract:**

Introduction: Durvalumab following chemoradiotherapy (CRT) for non-small cell lung cancer stage III has become the standard of care (SoC) in the past few years. With this regimen, 5-year overall survival (OS) has risen to 43%. Therefore, adequate pulmonary function (PF) after treatment is paramount in long-term survivors. In this respect, carbon monoxide diffusing capacity (DL_CO_), which represents the alveolar compartment, seems to be a suitable measure for residual lung capacity. The aim of the current analysis was to correlate DL_CO_ with pneumonitis and radiation dose. Patients and methods: One hundred and twelve patients with histologically confirmed NSCLC III treated between 2015/10 and 2022/03 were eligible for this study. Patients received two cycles of platinum-based induction chemotherapy followed by high-dose radiotherapy (RT). As of 2017/09, durvalumab maintenance therapy was administered for one year. The clinical endpoints were based on the thresholds jointly published by the European Respiratory Society (ERS) and the American Thoracic Society (ATS). Pre-treatment DL_CO_ of 60% was correlated to the incidence of pneumonitis, whereas the post-treatment DL_CO_ decline of 10% was related to radiation dose. Results: Patients with a pre-treatment DL_CO_ < 60% had a higher probability of pneumonitis (*n* = 98; r = 0.175; *p*-value 0.042), which could be reproduced in the subgroup of patients who did not receive durvalumab (*n* = 40; r = 0.288; *p*-value 0.036). In these individuals, the decline in DL_CO_ ≥ 10% depended significantly on the size of the lung volume receiving between 45% and 65% (V_65–45%_) of the total radiation dose (r = 0.354; *p*-value = 0.020) and V_20 Total Lung_ (r = 0.466; corrected *p*-value = 0.042). Conclusions: The current analysis revealed that DL_CO_ is a predictor for clinically relevant pneumonitis and a monitoring tool for post-treatment lung function as it correlates with radiation dose. This underlines the importance of peri-treatment lung function testing.

## 1. Introduction

Lung cancer is still one of the most prominent causes of cancer death worldwide [1]. The proportion of non-small cell histologies is around 80%, with about one-third of the patients presenting with UICC stage III disease. Locally advanced non-small cell lung cancer (LA-NSCLC) is a heterogeneous disease entity, which can be treated by surgery, radiotherapy (RT), chemotherapy, targeted drugs, and immunotherapy.

With the publication of the PACIFIC data in 2017 and its follow-up during the past five years [2,3,4], durvalumab after chemoradiation has become the standard of care (SoC) for NSCLC stage III patients [5]. The 5-year overall survival (OS) was 43% for all patients treated with durvalumab compared to 33% in the control arm [4]. This is a substantial improvement in long-term survival rates from below 20% during the last decades [6,7]. Integrating 18F-FDG-PET-CT in treatment planning and dose delivery by highly conformal radiation techniques such as intensity-modulated radiotherapy (IMRT) and volumetric arc therapy (VMAT) combined with immune checkpoint inhibition (ICI) with durvalumab are similarly responsible for this progress. Durvalumab is a high-affinity human IgG1 monoclonal antibody that prohibits binding between the programmed death receptor and its ligand. This allows T-cells to better recognize tumor cells, which leads to an enhanced immune response [3]. Like other ICIs, durvalumab may cause inflammation in general [8,9,10], leading to higher rates of pulmonary toxicity, i.e., pneumonitis. Although two recent publications on high-dose radiation together with durvalumab showed that the combination is safe [11,12], caution is still warranted. It is noteworthy that ICI maintenance therapy for one year after completion of chemoradiotherapy (CRT) increased the rate of any grade pulmonary toxicity to roughly 46.1% compared to the 31.2% in the placebo group [2,3].

In this context, post-treatment pulmonary function (PF) is especially important for long-term survivors. It has been argued that carbon monoxide diffusing capacity (DL_CO_) may be a better parameter for these lung function changes after RT than forced expiratory volume in one second (FEV_1_) as it represents the alveolar compartment [13]. In order to maintain a reasonable quality of life after thoracic chemo-radio-immunotherapy, post-treatment DL_CO_ should be at least 50% of the predicted reference value [14]. The European Respiratory Society (ERS), together with the American Thoracic Society (ATS), published a seminal paper on the cutoffs that allows distinguishing between normal and impaired lung function in clinical practice. According to this ESR/ATS guideline, a DL_CO_ < 60% and a decrease >10% compared to baseline can be regarded as clinically significant [15].

In order to properly assess pulmonary toxicity, it is relevant to know how far lung function correlates with CT morphology changes after treatment. In a previous publication by our group, we established a correlation between irradiated volumes and PF dynamics after high-dose radiation therapy. The volume receiving between 45% and 65% of the prescribed total radiation dose (V_65–45%_) was shown to be most prominently correlated with a DL_CO_ decline [16]. This is in line with the notion that radiation therapy may cause pulmonary fibrosis, which secondarily leads to PF impairment. At the same time, the widespread administration of ICI in lung cancer may complicate the predictability of post-treatment lung function based on radiation dosimetric parameters alone since durvalumab itself may cause pneumonitis [8,9,10].

Combining these previous results with the ERS/ATS thresholds, the current analysis pursues a two-fold purpose. First, we want to correlate pre-treatment PF with clinically relevant pneumonitis. Secondly, we intend to model post-treatment PF in dependence on radiation dose within a defined lung volume.

## 2. Methods

### 2.1. Patients

In the current analysis, 112 patients treated for inoperable NSCLC stage III with curative intent between 2015/10 and 2022/03 were included (approval by the ethics committee of the Federal State of Salzburg No. 415-E/1915/12-2015). All patients provided informed consent. The detailed selection criteria were described elsewhere [16]. In brief, patients with a suitable performance score (ECOG 0-1) were obligatorily staged with 18F-FDG-PET-CT and cranial MRI. They were followed up with thoracic contrast-enhanced CT and PF tests every three months for the first two years after completion of therapy and twice a year thereafter. For the current analysis, patients who had a minimum follow-up of one year were stratified according to whether or not they had received immunotherapy.

### 2.2. Sequential Chemo-Radio(-Immuno)Therapy

Following two cycles of platinum-based induction chemotherapy, all patients received sequential high-dose RT [17]. Two high-dose radiation treatment concepts were applied. Between 2015/10 and 2017/08, dose differentiated accelerated radiotherapy (DART) with two daily fractions of 1.8 Gy and total doses between 73.8 and 90.0 Gy in dependence on increasing tumor size was used [18]. As of 2017/09, an accelerated hypofractionated schedule with 3 Gy to the tumor up to a total dose of 66 Gy was implemented. This corresponds to the lowest dose level in the DART concept, with the biologically equivalent dose in 2 Gy fractions (EQD2) calculated according to the following formalism (*D* = total physical dose, *d* = single dose per fraction, *α/β* = 10 for tumor tissue).
(1)$$EQD2=D\frac{d+\alpha /\beta }{2+\alpha /\beta }$$

The dose constraints to the organs at risk were applied according to international standards [19]: the maximum dose (D_max_) to the spinal cord was 45 Gy. The limits for the mean esophageal dose (MED) and the volume of the heart receiving 25 Gy or more (V_25 Heart_) were set at 34 Gy and 10%, respectively [19]. The mean lung dose (MLD) had to be below 20 Gy, and the volume of both lungs receiving 20 Gy or more (V_20 Total Lung_) was below 40% [19]. As of September 2017, patients received durvalumab maintenance therapy for one year at a dose of 10 mg/kg within the Austrian early access program.

### 2.3. Clinical Endpoints

With the current SoC treatment for NSCLC UICC III, 5-year OS rates of 43% can be achieved [2,3,4]. In this context, both the probability of pulmonary toxicity after sequential high-dose CRT and lung function as a quality-of-life surrogate have become increasingly important. Hence the current study has two endpoints: clinically relevant pneumonitis, i.e., grade 2 or higher, and DL_CO_ decline of 10% or more.

#### 2.3.1. Pneumonitis

Pneumonitis scoring was based on CTCAE v5.0, comprised of respiratory symptoms such as cough and dyspnea, sometimes associated with fever as well as simultaneous radiological alterations in the follow-up CT scan and a reduced lung function compared to the latest PF test. The focus was placed on clinically relevant pulmonary toxicity starting from grade 2, which entails the administration of cortisone. Grade 1 was not assessed since most patients have asymptomatic radiographic changes in the lungs after high-dose sCRT. 

#### 2.3.2. DL_CO_ Thresholds

PF tests included whole-body plethysmography, blood gas analysis, and DL_CO_. In general, FEV_1_ and DL_CO_ are the most frequently used parameters to assess PF after chemo-radio-immunotherapy, with the latter being a surrogate marker for the alveolar compartment.

##### Pre-Treatment DL_CO_ Threshold of 60% 

In 2005, the ATS/ERS task force for the standardization of lung function testing published a seminal paper on thresholds for PF to distinguish normal from pathologic lung function. As for DL_CO_, values between 75% and 140% are regarded as normal, while decreases to 60–74%, 40–59%, and <40% are defined as mild, moderate, and severe, respectively.

##### Post-Treatment DL_CO_ Decline ≥10%

Respiratory medicine literature describes a DL_CO_ decline of 10% as clinically relevant [20,21,22] as far as it may have an impact on the patients’ quality of life. Therefore, we chose this cutoff as the clinical endpoint for modeling the correlation between radiation dose volumes and PF.

### 2.4. Modeling DL_CO_ as a Function of V_65–45%_

Based on the assumption that the anatomical substrate of CT density increase is fibrotic lung tissue that does not participate in gas exchange, we correlated DL_CO_ and CT morphology changes in a previous analysis [16]. Dose volumes with reference to the PTV were delineated on the planning CT in 10% decrements from V_105%_, which is the volume receiving at least 105% of the prescription dose, to V_5%_. This was the basis for creating differential volumes, e.g., V_105%-95%_ (=volume that received between 105% and 95% of the prescription dose). Lung density alterations in these differential volumes were related to DL_CO_. As a result, the most significant correlation was found with V_65–45%_. In the current paper, we intended to model the clinically relevant DL_CO_ decline > 10% as a function of the relative V_65–45%_ size. Hence, a binary data set for DL_CO_ decline after RT with a cutoff value of 10% was plotted against the relative size of V_65–45%_. The cumulative distribution function (CDF) was used for a probit fit to check whether V_65–45%_ could differentiate between a decline in DL_CO_ either below or above the 10% threshold using the minimum chi-square method. Assuming a normal distribution, the CDF can be described by the following formalism:(2)$$CDF\left(V_{rel}\right)=\frac{1}{2}\left(1+erf\left(\frac{V_{rel}-V_{Cutoff}}{\sigma \sqrt{2}}\right)\right)$$

*V_rel_* is the percentage of V_65–45%_ with respect to total lung volume (*V_rel_* = V_65–45%_/V_Lungs_ × 100), *V_Cutoff_* is the relative cutoff volume for the probability of developing a DL_CO_ decline ≥ 10%, *σ* is the standard deviation, and *erf* the error function. For significant differences in PF decline, a one-sided *t*-test was used, as we assumed that an improvement in DL_CO_ would be unexpected.

### 2.5. Statistics

OS and progression-free survival (PFS) were calculated with the Kaplan–Meier method. The comparison between subgroups of patients with or without ICI was performed with the logrank test. The correlation between DL_CO_, dosimetric, and clinical parameters was performed with the Pearson test with Bonferroni correction to adjust for multiple testing.

## 3. Results

### 3.1. Patients

With a median follow-up of 20.4 months (range: 5.9–66.5 months), 29/112 (26%) patients had died. The median age at baseline was 66 years (range: 29–85). The majority of the patients were male (67%), and almost all of them (96%) had a suitable performance score (ECOG 0–1). Although 63/112 (56%) patients had quit smoking. Still, about 1/3 were current smokers. Fifty-four patients (48%) had COPD but still a sufficient lung function to undergo sCRT with or without ICI (Table 1). The subgroups of patients stratified by immunotherapy did not differ significantly in baseline characteristics (Table A1).

### 3.2. Sequential Chemo-Radio(-Immuno)-Therapy 

All patients received two cycles of platinum-based induction chemotherapy combined with either gemcitabine or pemetrexed, depending on histology. After completion of RT, 64/112 patients (57%) received durvalumab maintenance therapy for one year (Table 2). The median gross tumor volume (GTV) was 29.4 mL (0.2–408 mL), with a median EQD2 to the tumor of 72.3 Gy (39–88.2). The median MLD was 12.0 Gy (4.0–18.0 Gy). By the end of 2018, VMAT was generally used as the technique of choice rather than step-and-shoot IMRT at our institute. In the wake of introducing durvalumab into clinical practice as of 2017/09, the radiation dose concept was modified to a moderate hypofractionation scheme with a median EQD2_Tumor_ of 71.3 Gy (range: 60.0–77.6 Gy) instead of the previous fierce dose escalation (EQD2_Tumor_ of 77.6; range: 58.3–88.2). This switch was based on the assumption that the trimodal treatment approach with durvalumab might be more effective as well as potentially more toxic. Hence, some of the technical treatment parameters differed significantly between subgroups (Table A2).

### 3.3. Clinical Outcome and Pulmonary Toxicity

The median follow-up was 20.4 months (range: 5.9–66.5). The Kaplan–Meier estimates for median OS and PFS were 57.9 months (95% CI: 44.5–71.3) and 33.9 months (95% CI: 15.0–52.8), respectively. While no difference between subgroups could be seen in terms of OS (Figure A1, logrank *p*-value = 0.357), PFS was significantly better for patients with ICI (Figure A2, 13.1 vs. 33.9 months, logrank *p*-value = 0.003). Sixteen cases (14%) of grade 2 or 3 pneumonitis occurred without clinically significant differences between subgroups (Table 3, Mann–Whitney U test *p*-value 0.691). 

### 3.4. Baseline DL_CO_ < 60% Predicts Pneumonitis

A total of 98 of the 112 patients (88%) had a DL_CO_ measurement before treatment. The median was 67% of the predicted reference value (range: 16–103%). Forty-one patients had a DL_CO_ below the 60% threshold, which is generally regarded as the cutoff between normal and mildly impaired DL_CO_ [15]. Of these, 9 patients (22%) developed pneumonitis, which is twice as high as in the 6/56 (11%) patients with baseline DL_CO_ > 60% (Table 4). We could establish a correlation between pre-therapeutic DL_CO_ (patients labeled according to whether their DL_CO_ was above or below the threshold) and the occurrence of pneumonitis (*n* = 98, one-sided Pearson correlation coefficient 0.175, *p*-value 0.042; Figure 1). While in the subgroup of patients without ICI, this correlation also reached statistical significance (*n* = 40, one-sided Pearson correlation coefficient 0.288; *p*-value 0.036), in those receiving ICI, it did not (*n* = 58, one-sided Pearson correlation coefficient 0.092, *p*-value 0.245) 

### 3.5. Short-Term DL_CO_ Decline > 10%

#### 3.5.1. Time Course of DL_CO_ Decline

Of the 112 patients (64 vs. 48 with and without ICI, respectively), 90 patients (56 vs. 34) had paired DL_CO_ measurements prior to as well as three months after RT. The median DL_CO_ decline was 9.5% (Q1 = 1.3%, Q3 = 17.5%) at three months compared to baseline, which was statistically significant (one-sided *t*-test *p* < 0.001). Of note, a “negative decline” is, in fact, an increase in DL_CO_. A decline larger than 10% was observed in 43/90 (47.8%) patients (Figure 2). In the subgroup of patients without ICI, the median decline was 11.3% (Q1 = 3.4%, Q3 = 17.5%, one-sided *t*-test, *p* < 0.001), with 18/34 patients (52.9%) above the threshold. Patients with ICI had a significant median decline of 8.1% (Q1 = −3.5%, Q3 = 16.2%, one-sided *t*-test, *p* < 0.001). A decline above 10% was observed in 25/56 (44.6%) patients. 

#### 3.5.2. DL_CO_ Decline ≥ 10% Correlates with V_65–45%_ > 5.1%

While in a previous study, we correlated CT morphology changes with DL_CO_ dynamics after RT [16], the current study shows that a clinically relevant DL_CO_ decline > 10% is significantly correlated with a higher percentage of the differential volume V_65–45%_. Figure 3 visualizes the relation between V_65–45%_ and a DL_CO_ decline > 10% three months after RT (one-sided Pearson correlation coefficient = 0.143, *p*-value = 0.089) with the cutoff point at 5.1%. This trend became significant in the subgroup of patients without ICI (Figure 4, one-sided Pearson correlation coefficient of 0.354, *p*-value = 0.020) but not for patients with ICI (Figure 5, one-sided Pearson correlation coefficient = −0.001, *p*-value = 0.498). As shown in Figure 4, V_65–45%_ above 5.1% entails a significantly higher probability for DL_CO_ decline ≥ 10%. The correlation between the size of V_65–45%_ and DL_CO_ decline > 10% in the patient subgroup that received two fractions per day was not significant (one-sided Pearson correlation coefficient = 0.205, *p*-value = 0.077). Therefore the influence of RT practice changes as a confounding factor seems to be weak. 

#### 3.5.3. DL_CO_ Decline Is Related to V_20 Total Lung_


Additionally, PF can be influenced by clinical and general therapeutic parameters. Hence, we correlated the following variables to DL_CO_ decline at three months both in the whole cohort and the subgroups separately: age at diagnosis, sex, pre-therapeutic weight loss, ECOG, smoking status, histology, UICC stage, COPD grade, GTV, tumor location, RT technique, the addition of immunotherapy, MLD, V_20 Total Lung_, EQD2_Tumor_, EQD2_Lymphnodes,_ and V_65–45%_. The only parameter that retained significance after Bonferroni correction was V_20 Total Lung_ in the non-ICI group. Patients with a higher V_20 Total Lung_ had a significantly higher probability for a DL_CO_ decline ≥ 10% (Table A3 and Figure A3: one-sided Pearson correlation coefficient = 0.466, corrected *p*-value = 0.042).

## 4. Discussion

The current analysis is based on internationally accepted DL_CO_ thresholds published by the ATS/ERS in 2005 [15]. In our cohort, the baseline DL_CO_ threshold of 60% predicts pneumonitis, and the DL_CO_ decline >10% correlates with dosimetric parameters such as the partial volume receiving 45% to 65% of the total dose (V_65–45%_) and the commonly used V_20 Total Lung_. Our findings corroborate previous investigations by our group, which revealed a strong correlation between CT morphology changes and the temporal dynamics of DL_CO_ after sCRT [16]. This underlines the clinical importance of PF tests as a prediction tool for toxicity as well as a monitoring parameter in clinical follow-up. Of note, in the subgroup of patients who received durvalumab, no such correlations could be found.

The patient population in the current analysis was comparable to prospective studies conducted in the field of NSCLC UICC III for the past two decades [2,3,4,6,7,23,24,25]. While the median PFS of 33.9 months in our cohort fits perfectly well into the current notion of improved disease control for patients receiving ICI [2,3,4], follow-up was not long enough to soundly judge OS. In accordance with the literature [2,3,4], it can be said that PFS is significantly longer for ICI patients (*p*-value = 0.003; Figure A2), and a trend toward better OS can be seen for this subcohort (*p*-value = 0.357; Figure A1). As for toxicity, the 14% pneumonitis observed in our cohort is substantially lower than in the 32% vs. 24% in the experimental and standard arms of the PACIFIC trial, respectively [2,3]. However, these crude numbers are not entirely comparable since we assessed clinically relevant pneumonitis requiring therapeutic intervention only.

As the 5-year survival rate for patients receiving durvalumab was 43% in the PACIFIC trial, CRT plus immunotherapy has become SoC [5]. For these long-term survivors, toxicity and lung function after treatment play a crucial role. This aspect is of even greater interest since, in the current analysis, sequential high-dose RT was combined with durvalumab, which potentially harbors an enhanced risk for pneumonitis. However, according to recent publications, it seems [11,12,26] that durvalumab can be safely applied together with sequential high-dose CRT without excess pulmonary toxicity [12]. Similarly, it has been shown that pulmonary toxicity after sequential high-dose CRT is in the range of cCRT with a 60 Gy total dose in 2 Gy fractions [18,27,28].

As mentioned above, patients with a DL_CO_ < 60% prior to RT had an increased probability of pneumonitis (Figure 1). While this difference became significant in the subgroup of patients without ICI, it was not seen in patients who had received durvalumab (Figure 1). Studies on DL_CO_ as a surrogate marker for lung function date back to the early 20th century [29]. Nowadays, according to ATS/ERS, a DL_CO_ between 75% and 140% represents normal global lung function, whereas 60–74% and 40–59% are mild and moderate decreases, respectively [15]. A decrease of 40% or below is considered severe [15]. From surgical [13,14,30,31] and radiation therapy studies [32,33,34], it is known that a pre-therapy DL_CO_ < 60% is associated with inferior outcomes. 

In a cohort of 98 patients, 60 of whom had UICC III, Brennan et al. found a sigmoid dose–response relationship between radiographic parameters, i.e., 4DCT ventilation data and PF. In this study, a cutoff >/<70% FEV_1_ was used to differentiate between normal and abnormal lung function [32]. Although Brennan et al. modeled a different PF parameter, we agree with the authors’ conclusion that advanced radiation treatment planning with VMAT/IMRT and online image guidance by cone-beam CT, together with PF monitoring, allow for the sparing of normal tissue in general and fibrotic areas specifically. 

In analogy to Brennan and colleagues, we also used a CT morphology parameter, i.e., V_65–45%_, to establish a sigmoid correlation with a PF test variable, i.e., DL_CO_ decline ≥ 10% (Figure 2), based on the ATS/ESR recommendations [15,20,22,35]. The median overall post-treatment decline in DL_CO_ of 10.5% is in the same range as in published reports [36]. Furthermore, in a previous investigation by our group, we could demonstrate a correlation between the temporal development of radiographic post-treatment changes and the dynamics of DL_CO_ in general. While in the above-mentioned investigation by our group, we could demonstrate a correlation between the temporal development of radiographic post-treatment changes and the dynamics of DL_CO_ [16], the current analysis revealed that V_65–45%_ higher than 5.1% is associated with a significantly increased probability in DL_CO_ decline of 10% or more. This finding could contribute to optimized radiotherapy planning by keeping V_65–45%_ below the 5% threshold in order to avoid PF loss after RT. The discrepancy between the statistically significant decrease in DL_CO_ in the subgroup of patients without ICI (Figure 4) but not in the durvalumab group (Figure 5) might be explained by the fact that ICI itself causes radiographic changes, i.e., fibrosis, which cannot be quantified by a model based on radiation dosimetry parameters only. Similar observations were reported by Borst et al., who also found a short-term decline in DL_CO_ three months after the end of RT in 34 patients. The decline persisted in the eight patients who were assessable 36 months after the end of RT, which might be important for long-term survivors [35]. The patient number, however, is very small, which precludes firm conclusions from this study. As for comparability with our data, it also has to be emphasized that the radiation techniques 20 years ago were different from current approaches to spare normal tissue. Moreover, patients did not receive ICI then.

In addition, Ma et al. also mentioned clinical and non-radiation-related treatment factors that may confound the correlation between CT morphology changes and PFTs [34]. For example, in central tumors, the shrinkage of the tumor leads to better lung ventilation, so there is the possibility of an inverse correlation between post-treatment PF tests and radiographic changes [34]. Furthermore, systemic treatments such as ICI may lead to pulmonary fibrosis [8,12], which is in line with the fact that we did not find a clear-cut correlation between radiation dosimetry and DL_CO_ decline in the subgroup of patients treated with durvalumab. Of note, in the non-immunotherapy group, this relation was highly significant, with a raw *p*-value of 0.003 (Bonferroni correction: 0.042) for V_20 Total Lung_ and DL_CO_. As this lung function parameter represents the alveolar compartment and is, therefore, a measure for gas exchange [13], it can be assumed that radiation-induced damage in the alveolar compartment reduces carbon monoxide diffusion across the alveolar-capillary membrane. Hence a larger V_65–45%_ or V_20 Total Lung_ entails a higher likelihood for DL_CO_ decline. In this context, it also has to be mentioned that V_65–45%_ and MLD were significant before Bonferroni correction for multiple testing. This means that these variables also have a certain predictive power, which is, however, markedly weaker compared to V_20 Total Lung_ (Table A3). Our findings corroborate a long-term study on 108 breast cancer patients, which revealed a highly significant correlation between V_20 Total Lung_ and DL_CO_ [37]. Although these long-term data are not entirely comparable to our study, the reported reduction of 14% is in the same range as in our analysis (Figure 1).

The current paper is limited by its retrospective nature, although the patients were collected in a prospective institutional registry. In spite of being one of the largest NSCLC UICC III cohorts in the field, we are aware that the number of patients is still too small to draw firm conclusions. A major limitation of the current study is the lack of patient-reported outcome data, which could be integrated with the objective PF results to gain a more precise notion of what a 10% DL_CO_ decline means in terms of daily quality of life. Finally, PF, in general, depends on a reproducible technical setup and patient compliance. The measurements for our study were conducted by the same team consisting of three technicians at the in-house pulmonology department. Inconsistencies due to patient positioning, as described by Katz [20], can be ruled out since the measurements were performed in a standardized sitting position.

## 5. Conclusions

In summary, we demonstrated that DL_CO_ is a predictor for pulmonary toxicity and a monitoring tool for post-treatment lung function, which corroborates the usefulness of peri-treatment lung function testing and helps to optimize radiation treatment planning. In the future, prospective studies in larger cohorts should integrate the pre-treatment DL_CO_ threshold of 60% as a parameter for patient selection in order to avoid pulmonary toxicity in patients with reduced pulmonary reserve. Additionally, V_65–45%_ may complement the range of dosimetric parameters that predict lung function after therapy.

## Figures and Tables

**Figure 1 cancers-15-01966-f001:**
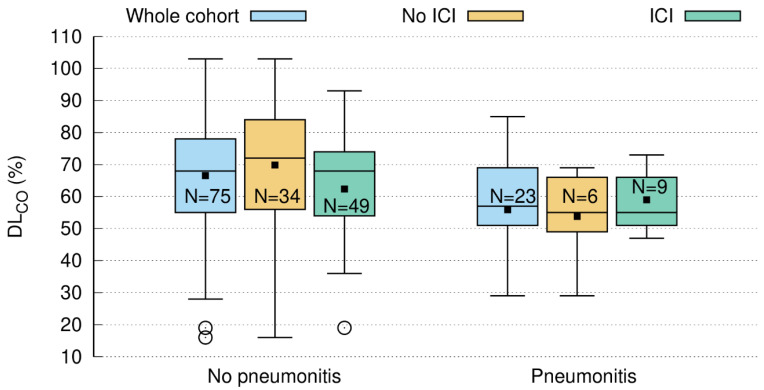
Pre-treatment DLCO (=carbon monoxide diffusing capacity) predicts the occurrence of clinically relevant pneumonitis, which was significant in the whole cohort (blue: *n* = 98; one-sided Pearson correlation coefficient 0.175; *p*-value = 0.042) and in the subcohort without ICI (= immune checkpoint inhibitor) shown in the brown boxplot (*n* = 40; one-sided Pearson correlation coefficient 0.288; *p*-value = 0.036). In patients with ICI (green), no such correlation could be found (*n* = 58; one-sided Pearson correlation coefficient 0.092; *p*-value = 0.245).

**Figure 2 cancers-15-01966-f002:**
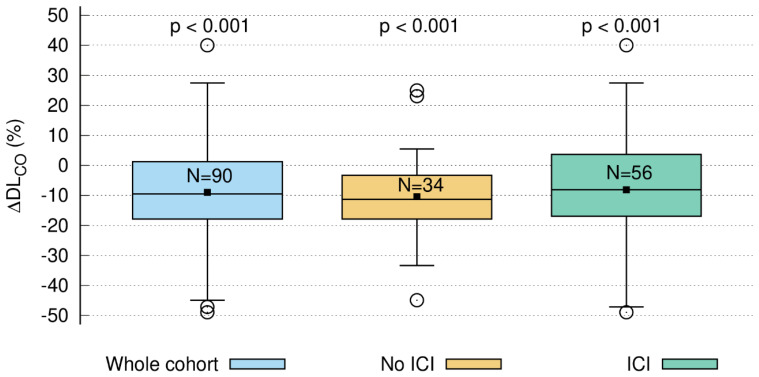
Relative decline in DLCO three months after radiotherapy compared to baseline (ΔDLCO): whole cohort (blue; *n* = 90), patients without ICI (=immune checkpoint inhibitor; brown; *n* = 34), and patients with ICI (green; *n* = 56). The decrease in DLCO (ΔDLCO) was significant for all three groups.

**Figure 3 cancers-15-01966-f003:**
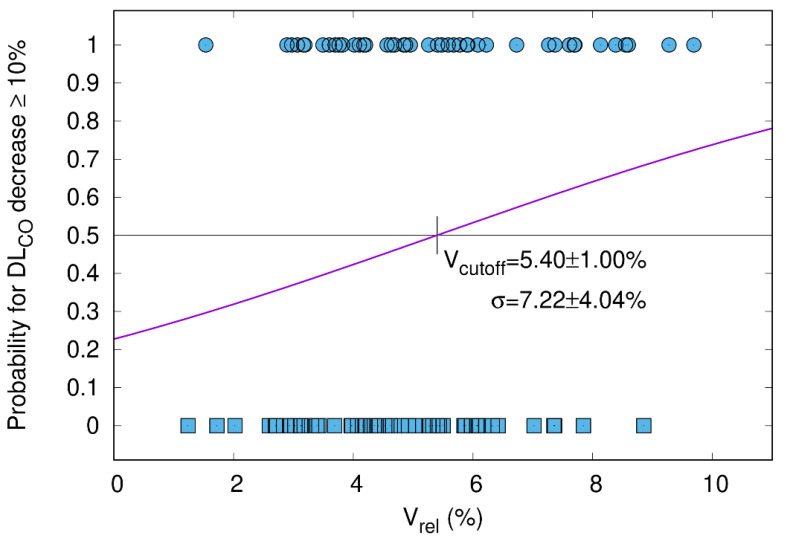
Whole cohort. The probability of DL_CO_ decrease ≥ 10% after the end of RT (y-axis) is plotted as a function of the relative size of V_65–45%_ with respect to total lung volume (V_rel_, x-axis). The sigmoid shape of the cumulative distribution function (CDF) shows that a larger V_65–45%_ entails a higher probability for DL_CO_ decline. All patients had paired measurements before and three months after radiotherapy (*n* = 90; Pearson correlation coefficient = 0.143; *p*-value = 0.089). The cutoff for V_rel_ was 5.4%. Patients with a DL_CO_ decrease of ≥10% and <10% are represented by circles and squares, respectively.

**Figure 4 cancers-15-01966-f004:**
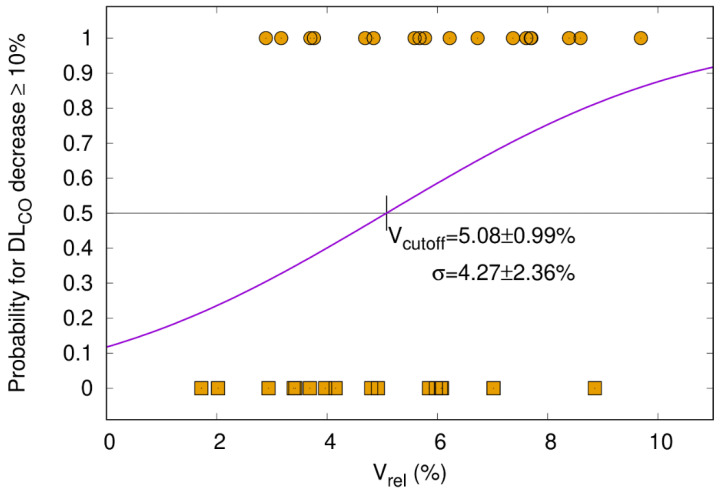
Patients without ICI (=immune checkpoint inhibitor). The probability of DL_CO_ decrease ≥ 10% after the end of RT (y-axis) is plotted as a function of the relative size of V_65–45%_ with respect to the total lung volume (V_rel_, x-axis). The sigmoid shape of the cumulative distribution function (CDF) shows that a larger V_65–45%_ entails a higher probability for DL_CO_ decline. Patients had paired measurements before and three months after radiotherapy (*n* = 34, Pearson correlation coefficient of 0.354, *p*-value = 0.020). The cutoff for V_rel_ was 5.1%. Patients with a DL_CO_ decrease of ≥10% and <10% are represented by circles and squares, respectively.

**Figure 5 cancers-15-01966-f005:**
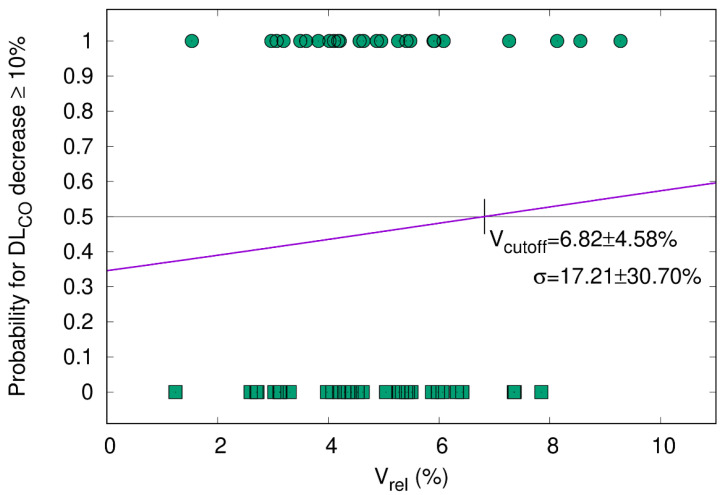
Patients with ICI (=immune checkpoint inhibitor). The probability of DL_CO_ decrease ≥ 10% after the end of RT (y-axis) is plotted as a function of the relative size of V_65–45%_ with respect to the total lung volume (V_rel_, x-axis). The cumulative distribution function (CDF) does not reveal a significant correlation. Patients had paired measurements before and three months after radiotherapy (*n* = 56, Pearson correlation coefficient = −0.001, *p*-value = 0.498). Patients with a DL_CO_ decrease of ≥10% and <10% are represented by circles and squares, respectively.

**Table 1 cancers-15-01966-t001:** Patient characteristics (NSCLC = non-small-cell lung cancer).

Patients *n* = 112		
Age (years)	Median	66
	Range	29–85
Sex	Male	75 (67%)
	Female	37 (33%)
Weight loss (%)	>5%	9 (8%)
	<5%	103 (92%)
ECOG	0–1	108 (96%)
	2	4 (4%)
Smoking status	Ex	63 (56%)
	Current	36 (32%)
	Never	11 (10%)
	Unknown	2 (2%)
Histology	NSCLC	112 (100%)
	Unknown	0 (0%)
UICC	III	112 (100%)
COPD grade	0	58 (52%)
	1	10 (9%)
	2	24 (21%)
	3	15 (13%)
	4	5 (5%)
	Unknown	0 (0%)

**Table 2 cancers-15-01966-t002:** Treatment (GTV = gross tumor volume, RT = radiotherapy, ICI = immune checkpoint inhibitor, MLD = mean lung dose, IMRT = intensity-modulated radiotherapy, VMAT = volumetric arc therapy, V20 Total Lung = lung volume receiving 20 Gy or more, EQD2 = biologically equivalent dose in 2 Gy fractions).

Treatment *n* = 112		
GTV (mL)	Median	29.4
	Range	0.3–408
Tumor location (*n*)	Peripheral	53 (47%)
	Central	59 (53%)
RT technique (*n*)	IMRT	52 (46%)
	VMAT	60 (54%)
ICI	Yes	64 (57%)
	No	48 (43%)
MLD (Gy)	Median	12.0
	Range	4.0–18.0
V_20 Total lung_ (%)	Median	20.0
	Range	5.3–37.0
EQD2 _Tumor_ (Gy)	Median	72.3
	Range	39.0–88.2
EQD2 _Lymphnodes_ (Gy)	Median	57.3
	Range	0–60.0

**Table 3 cancers-15-01966-t003:** The occurrence of clinically relevant pneumonitis, i.e., grades 2 to 5, did not differ significantly between subgroups with or without ICI (=immune checkpoint inhibitor; Mann–Whitney U test, *p*-value = 0.691).

Pneumonitis					
	Grade 2	Grade 3	Grade 4	Grade 5	*p*-Value
Whole cohort (*n* = 112)	13 (12%)	3 (3%)	0 (0%)	0 (0%)	n.a.
No ICI (*n* = 48)	4 (8%)	2 (4%)	0 (0%)	0 (0%)	0.691
ICI (*n* = 64)	9 (14%)	1 (2%)	0 (0%)	0 (0%)	

**Table 4 cancers-15-01966-t004:** Clinically relevant pneumonitis in dependence of baseline DL_CO_. If baseline DL_CO_ was <60%, the probability of clinically relevant pneumonitis was approximately twice as high as in patients with DL_CO_ above this threshold (n.a. = not assessable).

Pneumonitis in Dependence of Baseline DLCO			
	Baseline DLCO		
	<60%	>60%	n.a.
Pneumonitis cases	9 (22%)	6 (11%)	1

## Data Availability

The data presented in this study are available in this article.

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
