# Peer review of "Chemo-Radio-Immunotherapy for NSCLC III: ESR/ATS Thresholds for DLCO Correlate with Radiation Dosimetry and Pneumonitis Rate"

_cancers, 2023, doi:10.3390/cancers15071966_

Round 1

Reviewer 1 Report

This is an interesting paper that examined the association of baseline pulmonary function and the risk of pneumonitis, as well as post treatment change in DLCO with radiation dose to lung volumes, in an unselected cohort of patients with stage III NSCLC treated by chemoradiation. The authors report that the risk of pneumonitis was associated with a pretreatment DLCO < 60% and a decline in DLCO > 10% post treatment was associated with higher lung doses of radiation.

There are a number of aspects of the paper that could be improved as outlined below:

Introduction

·       While the implementation of durvalumab has been an important advance, I think the authors are overstating the impact. The authors repeatedly reference five year survival from PD-L1 positive patients in the PACIFIC trial. However, this is an unselected population of patients and not a cohort of PD-L1 positive patients. They should be referencing the overall trial results of 43% five year survival for durvalumab and 33% in the control arm ie only a 10% improvement in five year survival. They suggest that survival has gone from 20% to 50% because of immune checkpoint inhibition. However, the comparison is with studies at least 30 years old, with inferior radiation techniques and without PET staging. If they want to reference these old studies they need to put them in historical context.

·       Similarly the most appropriate estimate of pneumonitis rates in the era of checkpoint inhibition would come from the randomized PACIFIC trial where pneumonitis rates were 32% vs 24% in the control arm.

Methods

·       I think it is important to separate concepts of concurrent chemoradiation and sequential chemoradiation. There are a multitude of trials demonstrating superior survival for concurrent therapy. The methods appear to describe sequential chemoradiation. This terminology should be used throughout the article. This has relevance given the PACIFIC trial only included patients treated concurrently

·       The authors describe radiation dose in biological equivalents. They should specify how treatment was given in dose per fraction, number of fractions to better understand the relevance to usual treatment approaches of 60-66Gy in 30-33 fractions

·       How was the diagnosis of pneumonitis established. Were standardized criteria used and did this entail standardized imabing.

Results

·       Looking at the patient characteristics this is a good cohort of patients with the majority having minimal weight loss and many good PS patients.

·       If only 26% of patients have died, how can the median survival have been reached. Similarly with only a median follow up of only 20 months how can the median survival been nearly 5 years

·       The authors report the incidence of pneumonitis. They have dichotomised DLCO and it would be helpful to cross tabulate the number of cases of pneumonitis in patients with DLCO above and below 60%. This would enable to reader to better appreciate the discriminatory ability of the DLCO threshold

Discussion

·       Early in the paper, there is a comment about the importance of pulmonary function for long term survivors and quality of life. However, there is no reference to patient orientated outcomes such shortness of breath, or other respiratory symptoms. I think a big limitation in this study overall is the lack of correlation with patient reported outcomes. If the concern is long term toxicity with more patients living longer, then understanding this from the patient perspective would be important and the limitation acknowledged

·       The association of drop in DLCO >10% and radiation dose (V45%-65%) only appears in the cohort of patients who did not receive and immune checkpoint inhibitor. The authors suggest this might be because of pneumonitis associated with durvalumab. I think this is one of those examples of change in practice over time acting as a confounder. The later cohort of patients who received durvalumab, also received a lower biological equivalent dose of radiation and were treated with VMAT radiation techniques. These are very likely associated with the risk of radiation damage and drop in DLCO.

·       I believe that V20 is a more established predictor of pulmonary toxicity from radiation. The authors did not provide a strong rationale for using V45%-65% over V20. In the multivariate analysis, the only significant variable was V20. This warrants discussion, about the use of V20 versus V45%-65%

·       There is a paragraph on DLCO predicting non cancer mortality. This seems beyond the scope of the current study and no data is presented from the current study to support any of the assertions.

·       The limitations section needs expanding

·       It would be nice if the authors were able to tie back their findings to some future patient treatment considerations, or recommendations about next steps in research to better understand these issues

Author Response

Reviewer 1

This is an interesting paper that examined the association of baseline pulmonary function and the risk of pneumonitis, as well as post treatment change in DLCO with radiation dose to lung volumes, in an unselected cohort of patients with stage III NSCLC treated by chemoradiation. The authors report that the risk of pneumonitis was associated with a pretreatment DLCO < 60% and a decline in DLCO > 10% post treatment was associated with higher lung doses of radiation.

There are a number of aspects of the paper that could be improved as outlined below:

Introduction

  • While the implementation of durvalumab has been an important advance, I think the authors are overstating the impact. The authors repeatedly reference five year survival from PD-L1 positive patients in the PACIFIC trial. However, this is an unselected population of patients and not a cohort of PD-L1 positive patients. They should be referencing the overall trial results of 43% five year survival for durvalumab and 33% in the control arm ie only a 10% improvement in five year survival. They suggest that survival has gone from 20% to 50% because of immune checkpoint inhibition. However, the comparison is with studies at least 30 years old, with inferior radiation techniques and without PET staging. If they want to reference these old studies they need to put them in historical context.
  • Similarly the most appropriate estimate of pneumonitis rates in the era of checkpoint inhibition would come from the randomized PACIFIC trial where pneumonitis rates were 32% vs 24% in the control arm.

Methods

  • I think it is important to separate concepts of concurrent chemoradiation and sequential chemoradiation. There are a multitude of trials demonstrating superior survival for concurrent therapy. The methods appear to describe sequential chemoradiation. This terminology should be used throughout the article. This has relevance given the PACIFIC trial only included patients treated concurrently
  • The authors describe radiation dose in biological equivalents. They should specify how treatment was given in dose per fraction, number of fractions to better understand the relevance to usual treatment approaches of 60-66Gy in 30-33 fractions
  • How was the diagnosis of pneumonitis established. Were standardized criteria used and did this entail standardized imabing.

Results

  • Looking at the patient characteristics this is a good cohort of patients with the majority having minimal weight loss and many good PS patients.
  • If only 26% of patients have died, how can the median survival have been reached. Similarly with only a median follow up of only 20 months how can the median survival been nearly 5 years
  • The authors report the incidence of pneumonitis. They have dichotomised DLCO and it would be helpful to cross tabulate the number of cases of pneumonitis in patients with DLCO above and below 60%. This would enable to reader to better appreciate the discriminatory ability of the DLCO threshold

Discussion

  • Early in the paper, there is a comment about the importance of pulmonary function for long term survivors and quality of life. However, there is no reference to patient orientated outcomes such shortness of breath, or other respiratory symptoms. I think a big limitation in this study overall is the lack of correlation with patient reported outcomes. If the concern is long term toxicity with more patients living longer, then understanding this from the patient perspective would be important and the limitation acknowledged
  • The association of drop in DLCO >10% and radiation dose (V45%-65%) only appears in the cohort of patients who did not receive and immune checkpoint inhibitor. The authors suggest this might be because of pneumonitis associated with durvalumab. I think this is one of those examples of change in practice over time acting as a confounder. The later cohort of patients who received durvalumab, also received a lower biological equivalent dose of radiation and were treated with VMAT radiation techniques. These are very likely associated with the risk of radiation damage and drop in DLCO.
  • I believe that V20 is a more established predictor of pulmonary toxicity from radiation. The authors did not provide a strong rationale for using V45%-65% over V20. In the multivariate analysis, the only significant variable was V20. This warrants discussion, about the use of V20 versus V45%-65%
  • There is a paragraph on DLCO predicting non cancer mortality. This seems beyond the scope of the current study and no data is presented from the current study to support any of the assertions.
  • The limitations section needs expanding
  • It would be nice if the authors were able to tie back their findings to some future patient treatment considerations, or recommendations about next steps in research to better understand these issues

Submission Date

03 February 2023

Date of this review

14 Feb 2023 05:18:50

First of all, we would like to thank reviewer 1 for his/her in-depth analysis and the valuable comments on the manuscript, whose incorporation in the revised version has led to a substantial improvement. We provided point-by-point answers below (italics). The amended text that was integrated in the manuscript is in bold letters.

  1. While the implementation of durvalumab has been an important advance, I think the authors are overstating the impact. The authors repeatedly reference five year survival from PD-L1 positive patients in the PACIFIC trial. However, this is an unselected population of patients and not a cohort of PD-L1 positive patients. They should be referencing the overall trial results of 43% five year survival for durvalumab and 33% in the control arm ie only a 10% improvement in five year survival. They suggest that survival has gone from 20% to 50% because of immune checkpoint inhibition. However, the comparison is with studies at least 30 years old, with inferior radiation techniques and without PET staging. If they want to reference these old studies they need to put them in historical context.

Many thanks for this important remark. The following passages of the text were amended as suggested by the reviewer. The 43% OS for the whole PACIFIC cohort were quoted throughout the manuscript and the abstract as this – indeed - correlates better with the unselected patient population in our study. Additionally, it is true that the 20% were from two studies published in 1999 (Furuse) and 2011 (Curran), which we tried to put into historical perspective.

Introduction, 2nd paragraph

The 5-years overall survival (OS) was 43% for all patients treated with durvalumab compared to 33% in the control arm (Spigel). This is a substantial improvement in longterm survival rates from below 20% during the last decade (Furuse, Curran). Integrating 18F-FDG-PET-CT in treatment planning and dose delivery by highly conformal radiation techniques like intensity modulated radiotherapy (IMRT) and volumetric arc therapy (VMAT) combined with immune checkpoint inhibition (ICI) with Durvalumab are similarly responsible for this progress.

Methods, 2.3 Clinical endpoints

With the current standard of care (SoC) treatment for NSCLC UICC III, 5-years OS rates of 43% can be achieved (Antonia 2018, Antonia 2017, Spigel).

Discussion, 3rd paragraph

As the 5-year survival rate for patients receiving durvalumab was 43% in the PACIFIC trial (Spigel), CRT plus immunotherapy has become SoC (Ettinger). 

  1. Similarly the most appropriate estimate of pneumonitis rates in the era of checkpoint inhibition would come from the randomized PACIFIC trial where pneumonitis rates were 32% vs 24% in the control arm.

      We integrated the reviewer’s suggestion in the manuscript (please refer to discussion section, 2nd paragraph) emphasizing that the comparison between our study and the PACIFIC data has to be taken with a grain of salt since we did not assess grade 1 pneumonitis.

      As for toxicity, the 14% pneumonitis observed in our cohort are substantially lower than the 32% versus 24% in the experimental and standard arms of the PACIFIC trial, respectively (SJ Antonia 2017). However, these crude numbers are not entirely comparable with the current analysis since we assessed clinically relevant pneumonitis requiring therapeutic intervention only.

  1. I think it is important to separate concepts of concurrent chemoradiation and sequential chemoradiation. There are a multitude of trials demonstrating superior survival for concurrent therapy. The methods appear to describe sequential chemoradiation. This terminology should be used throughout the article. This has relevance given the PACIFIC trial only included patients treated concurrently.

This is an important point. We apologize for being unclear in this respect. According to the reviewer’s suggestion we clarified this throughout the manuscript, tables and legends using the term sequential chemoradiotherapy (sCRT) when refering to our cohort: sections 2.2, 2.3, 2.3.1, 3.1, 3.2, Discussion paragraphs 1 and 3, Tables A1 and A2.

  1. The authors describe radiation dose in biological equivalents. They should specify how treatment was given in dose per fraction, number of fractions to better understand the relevance to usual treatment approaches of 60-66Gy in 30-33 fractions.

      Many thanks for bringing up this aspect, which certainly helps to better clarify the radiation treatment concept. Hence we added the following sentences to the methods section:

      Two high dose radiation treatment concepts were applied. Between 2015/10 and 2017/08 dose differentiated accelerated radiotherapy (DART) with two daily fractions of 1.8 Gy and total doses between 73.8 to 90.0 Gy in dependence of increasing tumor size was used (Wurstbauer 2013). As of 2017/09 an accelerated hypofractioned schedule with 3 Gy to the tumor up to a total dose of 66 Gy was implemented. This corresponds to the lowest dose level in the DART concept, with the biologically equivalent dose in 2 Gy fractions (EQD2) calculated according to the following formalism (D total physical dose, d single dose per fraction, α/β = 10 for tumor tissue).

  1. How was the diagnosis of pneumonitis established? Were standardized criteria used and did this entail standardized imaging?

This – again – is a key issue, therefore we thank the reviewer to touch on it. As stated in section 2.3.1 we focused on clinically relevant pneumonitis, i.e. grade 2 or higher. Based on CTCAE v5.0 the criteria for the diagnosis of pneumonitis comprise respiratory symptoms (cough, dyspnea, sometimes associated with fever) together with radiomorphological alterations in the CT scan and a reduced lung function test. If – based on the clinical presentation, CT scan and the reduced lung function test – the  treating physician decided to administer cortisone, this condition was graded as pneumonitis grade 2 (administration of oxygen would be grade 3, admission to intensive care unit would be grade 4). We clarified this in section 2.3.1 Pneumonitis as follows.

Pneumonitis scoring was based on CTCAE v5.0 comprised respiratory symptoms such as cough and dyspnea sometimes associated with fever as well as simultaneous radiological alterations in the follow-up CT scan and a reduced lung function compared to the latest PF test.

  1. Looking at the patient characteristics this is a good cohort of patients with the majority having minimal weight loss and many good PS patients.

      Thank you for this remark.

  1. If only 26% of patients have died, how can the median survival have been reached? Similarly with only a median follow up of only 20 months how can the median survival been nearly 5 years?

      These numbers are the actuarial Kaplan-Meier estimates. We apologize for being unclear in this respect. The first sentence in section 3.3 Clinical outcome and pulmonary toxicity was adjusted as follows (please also see the table below containing the raw data, which were added for the sole purpose of answering the reviewers question):

The Kaplan-Meier estimates for median OS and PFS are 57.9 months (95%-CI: 44.5-71.3) and 33.9 months (95%-CI: 15.0-52.8), respectively.

We also want to stress that the data with respect to overall survival are not mature since the follow-up is too short. This is stated in the following sentence in second paragraph of the discussion:

While the median PFS of 33.9 months in our cohort fits perfectly well into the current notion of improved disease control for patients receiving ICI, follow-up was not long enough to soundly judge OS.

Overall survival

Zeit

Status

Kumulierter Anteil Überlebender zum Zeitpunkt

Anzahl der kumulativen Ereignisse

Anzahl der verbliebenen Fälle

Schätzer

Standardfehler

1

5,900

yes

,991

,009

1

111

2

7,300

no

.

.

1

110

3

8,200

no

.

.

1

109

4

8,300

yes

,982

,013

2

108

5

8,400

no

.

.

2

107

6

8,667

no

.

.

2

106

7

8,833

yes

,973

,016

3

105

8

8,900

no

.

.

3

104

9

9,300

no

.

.

3

103

10

9,733

no

.

.

3

102

11

9,800

no

.

.

3

101

12

10,000

yes

,963

,018

4

100

13

10,000

no

.

.

4

99

14

10,000

no

.

.

4

98

15

10,000

no

.

.

4

97

16

10,300

yes

,953

,020

5

96

17

10,333

no

.

.

5

95

18

10,500

no

.

.

5

94

19

11,000

yes

,943

,023

6

93

20

11,100

yes

,933

,025

7

92

21

11,300

no

.

.

7

91

22

11,400

yes

,923

,026

8

90

23

11,500

no

.

.

8

89

24

11,600

yes

,912

,028

9

88

25

11,700

no

.

.

9

87

26

11,767

no

.

.

9

86

27

11,967

no

.

.

9

85

28

12,100

no

.

.

9

84

29

12,200

yes

,901

,030

10

83

30

12,200

no

.

.

10

82

31

12,400

yes

.

.

11

81

32

12,400

yes

,879

,033

12

80

33

12,500

yes

,868

,034

13

79

34

12,900

no

.

.

13

78

35

12,900

no

.

.

13

77

36

13,100

no

.

.

13

76

37

13,600

no

.

.

13

75

38

13,800

no

.

.

13

74

39

14,100

no

.

.

13

73

40

14,200

no

.

.

13

72

41

14,200

no

.

.

13

71

42

14,567

no

.

.

13

70

43

15,200

no

.

.

13

69

44

15,800

yes

,856

,036

14

68

45

16,000

no

.

.

14

67

46

16,100

yes

,843

,038

15

66

47

16,400

no

.

.

15

65

48

16,700

yes

,830

,039

16

64

49

17,033

no

.

.

16

63

50

17,400

no

.

.

16

62

51

17,500

yes

,817

,041

17

61

52

18,000

no

.

.

17

60

53

18,000

no

.

.

17

59

54

18,300

yes

,803

,042

18

58

55

19,500

yes

,789

,044

19

57

56

20,300

no

.

.

19

56

57

20,500

no

.

.

19

55

58

20,800

no

.

.

19

54

59

21,333

no

.

.

19

53

60

21,567

no

.

.

19

52

61

21,600

no

.

.

19

51

62

21,600

no

.

.

19

50

63

22,300

no

.

.

19

49

64

22,433

no

.

.

19

48

65

22,700

no

.

.

19

47

66

23,300

no

.

.

19

46

67

24,133

no

.

.

19

45

68

24,400

yes

,771

,046

20

44

69

24,600

no

.

.

20

43

70

26,600

no

.

.

20

42

71

26,800

no

.

.

20

41

72

26,833

no

.

.

20

40

73

27,000

no

.

.

20

39

74

27,633

no

.

.

20

38

75

27,700

no

.

.

20

37

76

28,200

no

.

.

20

36

77

28,867

no

.

.

20

35

78

28,900

no

.

.

20

34

79

29,000

no

.

.

20

33

80

30,200

no

.

.

20

32

81

31,100

yes

,747

,051

21

31

82

31,200

no

.

.

21

30

83

31,500

yes

,722

,055

22

29

84

32,300

yes

,698

,058

23

28

85

32,300

no

.

.

23

27

86

33,400

yes

,672

,062

24

26

87

36,000

no

.

.

24

25

88

36,600

no

.

.

24

24

89

36,800

yes

,644

,065

25

23

90

38,300

no

.

.

25

22

91

39,800

no

.

.

25

21

92

41,800

no

.

.

25

20

93

42,100

yes

,612

,069

26

19

94

43,900

no

.

.

26

18

95

44,400

no

.

.

26

17

96

47,700

no

.

.

26

16

97

48,300

no

.

.

26

15

98

48,600

no

.

.

26

14

99

49,000

no

.

.

26

13

100

49,200

no

.

.

26

12

101

50,000

yes

,561

,080

27

11

102

52,000

no

.

.

27

10

103

53,300

no

.

.

27

9

104

56,200

no

.

.

27

8

105

57,900

yes

,491

,096

28

7

106

59,200

no

.

.

28

6

107

61,700

no

.

.

28

5

108

62,000

yes

,392

,117

29

4

109

62,700

no

.

.

29

3

110

62,800

no

.

.

29

2

111

65,500

no

.

.

29

1

112

66,500

no

.

.

29

0

Mean and median for overall survival

Mittelwerta

Median

Schätzer

Standardfehler

95%-Konfidenzintervall

Schätzer

Standardfehler

95%-Konfidenzintervall

Untergrenze

Obergrenze

Untergrenze

Obergrenze

47,588

2,760

42,178

52,999

57,900

6,848

44,478

71,322

a. Die Schätzung ist auf die längste Überlebenszeit begrenzt, wenn sie zensiert ist.

  1. The authors report the incidence of pneumonitis. They have dichotomised DLCO and it would be helpful to cross tabulate the number of cases of pneumonitis in patients with DLCO above and below 60%. This would enable the reader to better appreciate the discriminatory ability of the DLCO threshold.

      As suggested by reviewer, we added the following sentence in section 3.4 and Table 4 in the manuscript. It shows a 100% increase in the probability of clinically relevant pneumonitis in patients with baseline DLCO <60% compared to patients with baseline DLCO >60%.

      Of these, 9 patients (22%) developed pneumonitis, which is twice as high as in the 6/56 (11%) patients with baseline DLCO >60%.

      Table 4. Clinically relevant pneumonitis in dependence of baseline DLCO. If baseline DLCO was <60%, the probability of clinically relevant pneumonitis was approximately twice as high as in patients with DLCO above this threshold (n.a. = not assessible).

            Pneumonitis in dependence of baseline DLCO

Baseline DLCO

<60%

>60%

n.a.

Pneumonitis cases

9 (22%)

6 (11%)

1

  1. Early in the paper, there is a comment about the importance of pulmonary function for long term survivors and quality of life. However, there is no reference to patient orientated outcomes such shortness of breath, or other respiratory symptoms. I think a big limitation in this study overall is the lack of correlation with patient reported outcomes. If the concern is long term toxicity with more patients living longer, then understanding this from the patient perspective would be important and the limitation acknowledged.

      In fact, reviewer 1 is right in pointing out this major limitation. Hence we added the following sentence at the end of the discussion section (see also issue 13).

A major limitation of the current study is the lack of patient reported outcome data, which could be integrated with the objective PF results to gain a more precise notion of what a 10% DLCO decline means in terms of daily quality of life.

  1. The association of drop in DLCO >10% and radiation dose (V45%-65%) only appears in the cohort of patients who did not receive and immune checkpoint inhibitor. The authors suggest this might be because of pneumonitis associated with durvalumab. I think this is one of those examples of change in practice over time acting as a confounder. The later cohort of patients who received durvalumab, also received a lower biological equivalent dose of radiation and were treated with VMAT radiation techniques. These are very likely associated with the [reduced?] risk of radiation damage and drop in DLCO.

The reviewer’s concerns of “change in practice over time acting as a confounder“ sounds plausible. If we understand correctly, the reviewer assumes that patients who receive durvalumab, i.e. the patients that receive a lower total radiation dose delivered with VMAT, have a lower probability for a DLCO decline >10%.

In fact, the correlation coefficient between patients without immunotherapy and those being treated twice daily with a higher bioequivalent dose and a less conformal technique is r=0.667. However, when calculating the correlation coefficient between a decline in DLCO >10% and the size of V65%-45% in the respective groups, we find the following results. In the group without immunotherapy (N=34) the Pearson correlation coefficient is r = 0.354 (p = 0.020 for a one-sided analysis). In the subgroup with a higher bioequivalent dose and a less conformal technique (N=50) the correlation coefficient r is 0.205 (p = 0.076 for a one-sided analysis). Due to this difference in significance, we argue that the correlation effect between a decline in DLCO >10% and the size of V65%-45% is present in the subcohort without immunotherapy but not in the durvalumab group. For clarification we include the table below in our point-by-point answer to reviewer 1 (but not in the manuscript) and the following sentences at the end of section 3.5.2.

Correlation between V65%-45% and DLCO decline >10%

Patients without immunotherapy

r

0.354

N

34

Patients with twice daily fractions

r

0.205

N

50

The correlation between the size of V65%-45% and DLCO decline >10% in the patient subgroup that received two fractions per day was not significant (one-sided Pearson correlation coefficient = 0.205, p-value = 0.077). Therefore the influence of RT practice changes as a confounding factor seem to be weak.

  1. I believe that V20 is a more established predictor of pulmonary toxicity from radiation. The authors did not provide a strong rationale for using V45%-65% over V20. In the multivariate analysis, the only significant variable was V20. This warrants discussion, about the use of V20 versus V45%-65%.

      We agree with the reviewer that V20 is an established parameter to evaluate pulmonary toxicity caused by radiation. The intention of the current analysis was not to replace V20 by V65%-45%. This latter volumetric parameter was the result of a previous analysis by our group showing a strong correlation between DLCO dynamics and radiographic changes in the area of the lung, which received between 65% and 45% of the prescribed dose (V65%-45%; cf. Stana et al. doi: 10.3390/diagnostics12051027).

      In accordance with the reviewer’s remark we also included V65%-45% in the correlation analysis shown in Table A3 together with well established dosimetric parameters and clinical features. With a p-value of 0.020 V65%-45% also showed a significance level in the range of MLD (p-value = 0.010), yet much weaker than V20 (raw p-value 0.003). Therefore, after Bonferroni correction V 65%-45% - similar to MLD – was no longer significant. In conclusion, one might argue that the predictive value of V65%-45% is similar to MLD but much weaker than for V20. This new Table A3 entailed changes in the manuscript text (Results section 3.5.3, Discussion last but one paragraph).

Table A3. Correlation of DLCO decline >10% with clinical and treatment related parameters (GTV = gross tumor volume, ICI = immune checkpoint inhibitor, MLD = mean lung dose, V20 Total Lung = Lung volume receiving 20 Gy or more, EQD2 = biologically equivalent dose in 2 Gy fractions), V65%-45% (Lung volume receiving between 65% and 45% of the prescribed dose). Bonferroni correction was performed to account for multiple testing.

Pearson correlation coefficient

raw p-value

corrected p-value

Age

0.099

0.290

n.s.

Sex

0.022

0.450

n.s.

Weight loss

-0.274

0.058

n.s.

ECOG

-0.204

0.123

n.s.

Smoking status

0.038

0.416

n.s.

Histology

n.a.

UICC stage

n.a.

COPD

0.038

0.415

n.s.

GTV

0.060

0.367

n.s.

Tumor location

-0.174

0.163

n.s.

Radiation technique

-0.122

0.246

n.s.

ICI

n.a.

MLD

0.394

0.010

n.s.

V20 Total Lung

0.466

0.003

0.042

EQD2 Tumor

0.224

0.102

n.s.

EQD2 Lymphnodes

0.257

0.071

n.s.

V65%-45%

0.354

0.020

n.s.

Results, section 3.5.3

Hence, we correlated the following variables to DLCO decline at three months both in the whole cohort and the subgroups separately: age at diagnosis, sex, pre-therapeutic weight loss, ECOG, smoking status, histology, UICC stage, COPD grade, GTV, tumor location, RT technique, addition of immunotherapy, MLD, V20 Total Lung, EQD2Tumor, EQD2Lymphnodes and V65%-45%.

      Discussion, last but one paragraph

      In this context it also has to be mentioned that V65%-45% and MLD were significant before Bonferroni correction for multiple testing. This means that these variables also have a certain predictive power, which is – however – markedly weaker compared to V20 (Table A3).

  1. There is a paragraph on DLCO predicting non cancer mortality. This seems beyond the scope of the current study and no data is presented from the current study to support any of the assertions.

      On re-evaluation we agree with reviewer 1. Consequently we deleted the following part of the 4th paragraph in the discussion section.

      In a slightly smaller cohort than ours (68 stage III patients) Liptay et al. showed that the mean DLCO of patients who died from causes other than cancer was 54%13. Multivariate analysis revealed that this PF was highly significantly associated with non-cancer mortality. This underlines the notion that a pre-treatment DLCO <60% is associated with higher non-cancer mortality 13. One of the reasons may be that DLCO is a better indicator for “cardiopulmonary reserve”, while FEV1 is a good marker for obstruction 32. DLCO represents the alveolar compartment as it is a measure for gas exchange across the alveolar-capillary membrane 13. Therefore - in accordance with studies on combined pulmonary fibrosis and emphysema (CPFE) 33,34 – it is not counterintuitive to regard pneumonitis as a combination of radiation-induced fibrosis and pre-existing emphysema based on COPD, especially when taking into consideration that 46% of the patient population in our study had COPD 1 to 4 (Table 1).

  1. The limitations section needs expanding.

As requested, we enlarged the limitations section by adding the following sentences (please also see issue 9).

A major limitation of the current study is the lack of patient reported outcome data, which could be integrated with the objective PF results to gain a more precise notion of what a 10% DLCO decline means in terms of daily quality of life.

  1. It would be nice if the authors were able to tie back their findings to some future patient treatment considerations, or recommendations about next steps in research to better understand these issues.

Thank you for this important remark. As a consequence we modified the conclusion section as follows:

In the future, prospective studies in larger cohorts should integrate the pre-treatment DLCO  threshold of 60% as a parameter for patient selection in order to avoid pulmonary toxicity in patients with reduced pulmonary reserve. Additionally, V65%-45% may complement the range of dosimetric parameters that predict lung function after therapy.

Reviewer 2 Report

The manuscript by Stana et al. reported the analysis of correlation between the carbon monoxide diffusing capacity (DLCO) of lung and the incidence of pneumonitis, radiation dosimetry during the Chemo-radio-immunotherapy. DLCO is one of the clinically most widespread used lung function parameters. Their analysis showed pre-treatment DLCO of 60% was correlated to the incidence of pneumonitis. Furthermore, the post-treatment DLCO decline of 10% depended significantly on the size of the lung volume receiving between 45% and 65% of the total radiation dose and on the size of the lung volume receiving radiation more than 20 Gy. Overall, I found their analysis is interesting and underlines the importance of peri-treatment lung function testing. My main concerns are the writing and methods section.

1.       For the analysis between DLCO and radiation dosimetry, the authors didn’t clearly show the rationale behind the analysis. Why the authors did this? In the discussion part, the authors should discuss why DLCO decline correlates with V65%-45%, and V20 Total Lung.

2.       The manuscript incorporates the immunotherapy. The authors should discuss the related results in the discussion part.

3.       In the methods section, the authors should add the CT Morphology part, which may help the reads to follow the concept of V65%-45%.

4.       A lot of abbreviations are not with the full name, which should be checked.

5.       The Keywords of the manuscript are not in line with the content

Author Response

Reviewer 2

The manuscript by Stana et al. reported the analysis of correlation between the carbon monoxide diffusing capacity (DLCO) of lung and the incidence of pneumonitis, radiation dosimetry during the Chemo-radio-immunotherapy. DLCO is one of the clinically most widespread used lung function parameters. Their analysis showed pre-treatment DLCO of 60% was correlated to the incidence of pneumonitis. Furthermore, the post-treatment DLCO decline of 10% depended significantly on the size of the lung volume receiving between 45% and 65% of the total radiation dose and on the size of the lung volume receiving radiation more than 20 Gy. Overall, I found their analysis is interesting and underlines the importance of peri-treatment lung function testing. My main concerns are the writing and methods section.

  1. For the analysis between DLCO and radiation dosimetry, the authors didn’t clearly show the rationale behind the analysis. Why the authors did this? In the discussion part, the authors should discuss why DLCO decline correlates with V65%-45%, and V20 Total Lung.
  2. The manuscript incorporates the immunotherapy. The authors should discuss the related results in the discussion part.
  3. In the methods section, the authors should add the CT Morphology part, which may help the reads to follow the concept of V65%-45%.
  4. A lot of abbreviations are not with the full name, which should be checked.
  5. The Keywords of the manuscript are not in line with the content

First of all, we would like to thank reviewer 2 for her/his thorough analysis and valuable comments on the manuscript. Please see our point-by-point answers below (italics) and the insertions in the manuscript text (bold).

  1. For the analysis between DLCO and radiation dosimetry, the authors didn’t clearly show the rationale behind the analysis. Why the authors did this? In the discussion part, the authors should discuss why DLCO decline correlates with V65%-45%, and V20 Total Lung.

      DLCO as a pulmonary function parameter represents gas exchange (Pellegrino et al. doi: 10.1183/09031936.05.00035205; Liptay et al. doi: 10.1002/jso.21407; Katz et al. doi: 10.1186/s12890-018-0723-4). In a previous analysis by our group (cf. Stana et al. doi: 10.3390/diagnostics12051027) we could demonstrate a strong correlation between the radiographic changes in the mid-dose range, especially in the lung volume that received between 45% and 65% of the radiation dose (V65%-45%) and DLCO. Building upon this previous analysis, the current study focuses on the correlation of these findings with ERS/ATS thresholds for DLCO. Both aspects (i.e. results of the previous study and the purpose / rationale of the current analysis) are stated at the end of the introduction section. We apologize for being unclear with respect to the discussion section, which was amended according to the reviewer’s suggestion (Discussion, last but one paragraph):

      Introduction, last but one paragraph:

      In a previous publication by our group, we established a correlation between irradiated volumes and PF dynamics after high dose radiation therapy. The volume receiving between 45% and 65% of the prescribed total radiation dose (V65%-45%) was shown to be most prominently correlated with a DLCO decline (Stana 2022).

      Introduction, last paragraph:

      Combining these previous results with the ERS/ATS thresholds, the current analysis pursues a two-fold purpose. First, we want to correlate pre-treatment PF with clinically relevant pneumonitis. Secondly, we intend to model post-treatment PF in dependence of radiation dose within a defined lung volume.

      Discussion, last but one paragraph:

      As this lung function parameter represents the alveolar compartment and is therefore a measure for gas exchange (Liptay), it can be assumed that a radiation induced damage in the alveolar compartment reduces carbon monoxide diffusion across the alveolar-capillary membrane. Hence a larger proportion of the lungs receiving 45% to 65% of the prescribed dose (V65%-45%) or ≥20 Gy (V20) entails a higher likelihood for DLCO decline.           

  1. The manuscript incorporates the immunotherapy. The authors should discuss the related results in the discussion part.

      We apologize for being imprecise in this respect. The results for immunotherapy patients are mentioned in paragraphs 2,4 and 6 of the discussion section. We inserted the following sentences in paragraph 2 to clarify this important issue raised by the reviewer. As for ESR/ATS thresholds we would like to kindly ask the reviewer to refer to paragraphs 4 and 6, in which the results are discussed for non-ICI and ICI subcohort separately. While the first refers to pneumonitis, the latter summarizes the results on DLCO decline.  

      In accordance with literature (Antonia 2017, Antonia 2018, Spigel 2022), it can be said that PFS is significantly longer for ICI patients (p-value = 0.003; Figure A2) and a trend towards better OS can be seen for this subcohort (p-value = 0.357; Figure A1).

      (…)

      As mentioned above, patients with a DLCO <60% prior to RT had an increased probability for pneumonitis (Figure 1). While this difference became significant in the subgroup of patients without ICI, it was not seen in patients who had received durvalumab (Figure 1).

      (…)

      The discrepancy between the statistically significant decrease in DLCO in the subgroup of patients without ICI (Figure 3), but not in the Durvalumab group (Figure 4) might be explained by the fact that ICI itself causes radiographic changes, i.e. fibrosis, which cannot be quantified by a model based on radiation dosimetry parameters only.

  1. In the methods section, the authors should add the CT Morphology part, which may help the reads to follow the concept of V65%-45%.

      Many thanks for this important remark, we added the following paragraph in the methods section 2.4 Modelling DLCO as a function of V65%-45%.

      Based on the assumption that the anatomical substrate of CT density increase is fibrotic lung tissue that does not participate in gas exchange, we correlated DLCO and CT morphology changes in a previous analysis (Stana 2022). Dose volumes with reference to the PTV were delineated on the planning CT in 10% decrements from V105% (= volume receiving at least 105% of the prescription dose), to V5%. This was the basis to create differential volumes, e.g. V105%-95% (= volume that received between 105% and 95% of the prescription dose). Lung density alterations in these differential volumes were related to DLCO. As a result, the most significant correlation was found with V65%-45%.

  1. A lot of abbreviations are not with the full name, which should be checked.

      We apologize for this inaccuracy. Abbreviations are explained when appearing first in the manuscript and listed in the abbreviations sections at the end of the main text after Conclusions.

  1. The Keywords of the manuscript are not in line with the content.

      This is a terrible mistake, which we have no explanation for. The correct key words are listed below:

      NSCLC stage III, Durvalumab, immunotherapy, chemo-radiotherapy, ESR/ATS thresholds, DLCO

Reviewer 3 Report

The authors report the incidence of pneumonitis correlation between carbon monoxide diffusing capacity and radiation dosimetry in 112 patients with stage III non-small cell lung cancer. Pulmonary capacity evaluation after chemotherapy and radiotherapy is critical for monitoring treatment response and disease progression, improving patient quality of life and long-term survival. The study finds that pre-treatment patients have a high chance of developing pneumonitis when carbon monoxide diffusing capacity is less than 60%. In addition, in posttreatment patients, an over 10% reduction in carbon monoxide diffusing capacity is closely associated with the size of the lung volume receiving between 45% and 65% of the total radiation dose. Carbon monoxide diffusing capacity may serve as a good marker for predicting clinically relevant pneumonitis and assessing posttreatment lung function, while the sample size is relatively small with the study performed in a single study. I recommend the manuscript for publication in Cancers after the authors address the following issues:

1.       Please provide a patient consent statement in the manuscript.

2.       In the manuscript, the authors state that durvalumab can be safely applied together with sequential high-dose CRT without excess pulmonary toxicity. How about the chemotherapeutic agents? Would the chemotherapy associate potential pulmonary toxicity and confound the carbon monoxide diffusing capacity since radiation dosimetry is the critical predictor discussed here?

3.       Please provide full names for all abbreviations during their first appearances, such as COPD and 4DCT.

4.       Please check the references, such as refs, 4, 9, 29, to ensure the format is correct.

Author Response

Reviewer 3

The authors report the incidence of pneumonitis correlation between carbon monoxide diffusing capacity and radiation dosimetry in 112 patients with stage III non-small cell lung cancer. Pulmonary capacity evaluation after chemotherapy and radiotherapy is critical for monitoring treatment response and disease progression, improving patient quality of life and long-term survival. The study finds that pre-treatment patients have a high chance of developing pneumonitis when carbon monoxide diffusing capacity is less than 60%. In addition, in posttreatment patients, an over 10% reduction in carbon monoxide diffusing capacity is closely associated with the size of the lung volume receiving between 45% and 65% of the total radiation dose. Carbon monoxide diffusing capacity may serve as a good marker for predicting clinically relevant pneumonitis and assessing posttreatment lung function, while the sample size is relatively small with the study performed in a single study. I recommend the manuscript for publication in Cancers after the authors address the following issues:

  1. Please provide a patient consent statement in the manuscript.
  2. In the manuscript, the authors state that durvalumab can be safely applied together with sequential high-dose CRT without excess pulmonary toxicity. How about the chemotherapeutic agents? Would the chemotherapy associate potential pulmonary toxicity and confound the carbon monoxide diffusing capacity since radiation dosimetry is the critical predictor discussed here?
  3. Please provide full names for all abbreviations during their first appearances, such as COPD and 4DCT.
  4. Please check the references, such as refs, 4, 9, 29, to ensure the format is correct.

We would like to thank reviewer 3 for his/her remarks. Please find our point-by-point answers below (italics) and the respective alterations in the manuscript text (bold).

  1. Please provide a patient consent statement in the manuscript.

We apologize for this carelessness. The statement was provided in section 2.1 Patients.

All patients provided informed consent.

  1. In the manuscript, the authors state that durvalumab can be safely applied together with sequential high-dose CRT without excess pulmonary toxicity. How about the chemotherapeutic agents? Would the chemotherapy associate potential pulmonary toxicity and confound the carbon monoxide diffusing capacity since radiation dosimetry is the critical predictor discussed here?

We thank reviewer 3 for pointing out this important issue. In order to clarify it we added the following sentence to the discussion section, at the end of the third paragraph.

Similarly, it has been shown that pulmonary toxicity after sequential high-dose CRT is in the range of cCRT with 60 Gy total dose in 2 Gy fractions (Wurstbauer 2017, Auperin 2010, Kuang 2022).

  1. Please provide full names for all abbreviations during their first appearances, such as COPD and 4DCT.

      We apologize for being in accurate. As requested by the reviewer, abbreviations were explained at first appearance and summarized at the end of the manuscript.

  1. Please check the references, such as refs, 4, 9, 29, to ensure the format is correct.

The following references were re-checked with endnote and inserted in the correct form as given below: ref 4 = Spigel DR et al., ref 10 = Fukihara J et al., ref 34 = Ma J et al.

Round 2

Reviewer 1 Report

This is a re review of previously submitted manuscript. The authors have responded to the previous comments. There are still limitations to the manuscript but these are more to do with the nature of the study design.

My one remaining question relates to the utility of V45-65%. The authors describe a correlation between decline in DLCO >10% and V45-65%. While this is descriptive, unlike V20, it does not help the treating physician when evaluating an individual patient. As an example, V20 should be kept below 35%. Is there a threshold value for V45-65% that would exceed radiation treatment parameters. This would provide some predictive value to this radiation parameter. 

Author Response

My one remaining question relates to the utility of V45-65%. The authors describe a correlation between decline in DLCO >10% and V45-65%. While this is descriptive, unlike V20, it does not help the treating physician when evaluating an individual patient. As an example, V20 should be kept below 35%. Is there a threshold value for V45-65% that would exceed radiation treatment parameters. This would provide some predictive value to this radiation parameter. 

Submission Date

03 February 2023

Date of this review

20 Mar 2023 16:41:06

Again, we thank reviewer 1 for this pertinent remark refering to the utility of V65%-45%. Please refer to our direct answers (italics) and insertions in the manuscript text (bold) below.

In a previous publication by our group (cf. Stana et al. doi: 10.3390/diagnostics12051027), we  showed that the tissue density changes in the volume receiving 45%-65% of the total dose (V65%-45%)  were highly significantly related to the dynamics of DLCO. In other words, the radiographic changes in follow-up CTs after radiotherapy specifically in this volume represent anatomically the physiological changes in pulmonary function. As shown in figure 4, the cutoff for V65%-45% was 5.1%, which means that patients with a V65%-45% above this threshold have a significantly higher probability for DLCO decline of 10% or more after treatment. We emphasized this by changing the subheading in the results section and adding the following sentences in  Results and Discussion stressing the utility of this parameter for radiation treatment planning.

3.5.2 DLCO decline ≥10% correlates with V65%-45% >5.1%

As shown in figure 4, V65%-45% above 5.1% entails a significantly higher probability for DLCO decline ≥10%.

  1. Discussion

While in the above mentioned investigation by our group we could demonstrate a correlation between the temporal development of radiographic post-treatment changes and the dynamics of DLCO (Stana et al. doi: 10.3390/diagnostics12051027), the current analysis revealed that V65%-45% higher than 5.1% is associated with a significantly increased probability in DLCO decline of 10% or more. This finding could contribute to optimized radiotherapy planning by keeping V65%-45% below the 5%-threshold in order to avoid PF loss after RT.

Reviewer 2 Report

I appreciate the efforts made by authors to improve the manuscript. I have no more questions.

Author Response

Reviewer 2

I appreciate the efforts made by authors to improve the manuscript. I have no more questions.

Many thanks for this remark!